# Germline Genetic Mutations in Adult Patients with Sarcoma: Insight into the Middle East Genetic Landscape

**DOI:** 10.3390/cancers16091668

**Published:** 2024-04-25

**Authors:** Ramiz Abu-Hijlih, Baha Sharaf, Samer Salah, Hira Bani Hani, Mohammad Alqaisieh, Abdulla Alzibdeh, Layan Ababneh, Suleiman Mahafdah, Hikmat Abdel-Razeq

**Affiliations:** 1Department of Radiation Oncology, King Hussein Cancer Center, Amman 11941, Jordan; rhijlih@khcc.jo (R.A.-H.); aa.15218@khcc.jo (A.A.); 2Department of Internal Medicine, King Hussein Cancer Center, Amman 11941, Jordan; bs.13628@khcc.jo (B.S.); ds.06907@khcc.jo (S.S.); hb.14507@khcc.jo (H.B.H.); malqaisieh@hhcs.org (M.A.); 3School of Medicine, Jordan University of Science and Technology, Irbid 22110, Jordan; laababneh185@med.just.edu.jo; 4Department of Surgery, Royal Jordanian Medical Services, Amman 11855, Jordan; sleman.mahafza.sm@gmail.com; 5School of Medicine, The University of Jordan, Amman 11942, Jordan

**Keywords:** sarcoma, osteosarcoma, soft tissue sarcoma, pathogenic germline variants

## Abstract

**Simple Summary:**

Although the majority of sarcoma cases are sporadic, some are linked to familial and genetic predisposition phenotypes. Data on germline genetic testing for sarcoma are scarce, despite its increasing utility in enabling the selection of therapeutic options, cancer screening, and familial testing and counseling. Genetic germline testing (GGT) is recommended for specific subtypes of sarcoma; nevertheless, currently there are no uniform guidelines to guide GGT in sarcoma patients. In this prospective study, we investigated newly diagnosed patients with sarcoma to better understand the landscape of pathogenic genetic variants (PGVs) in our region and explore the potential actionability of these alterations. Out of 87 enrolled patients, 18 (20.7%) had PGVs. Younger age, presence of a second primary tumor, and female gender were significantly associated with higher PGV rates. The majority of detected mutations were potentially actionable and almost all mutations had implications on cancer screening and family counselling.

**Abstract:**

Data on germline mutations in soft tissue and bone sarcomas are scarce. We sought to identify the prevalence of germline mutations in adult sarcoma patients treated at a tertiary cancer center. Newly diagnosed patients were offered germline genetic testing via an 84-gene panel. The prevalence of pathogenic germline variants (PGVs) and their association with disease-, and patient- related factors are reported. A total of 87 patients were enrolled, the median age was 48 (19–78) years, and 47 (54%) were females. Gastrointestinal stromal tumors (*n* = 12, 13.8%), liposarcoma (*n* = 10, 11.5%), and Ewing sarcoma (*n* = 10, 11.5%) were the main subtypes. A total of 20 PGVs were detected in 18 (20.7%) patients. Variants of uncertain significance, in the absence of PGVs, were detected in 40 (45.9%) patients. Young age (*p* = 0.031), presence of a second primary cancer (*p* = 0.019), and female gender (*p* = 0.042) were correlated with the presence of PGVs. All identified PGVs have potential clinical actionability and cascade testing, and eight (44.44%) suggested eligibility for a targeted therapy. Almost one in five adult patients with soft tissue and bone sarcomas harbor pathogenic or likely pathogenic variants. Many of these variants are potentially actionable, and almost all have implications on cancer screening and family counselling. In this cohort from the Middle East, younger age, presence of a second primary tumor, and female gender were significantly associated with higher PGVs rates. Larger studies able to correlate treatment outcomes with genetic variants are highly needed.

## 1. Introduction

Sarcomas are rare malignant neoplasms of mesenchymal origin that account for 1–2% of all adult cancers and approximately 10% of childhood and adolescence cancers [1,2]. Although broadly divided into soft tissue sarcomas (STSs) and bone sarcomas, these tumors encompass a wide spectrum of histologic subtypes, with around 100 distinct subtypes currently identified [3]. Molecular testing, in addition to conventional morphologic and histochemistry tests, performed on tumor tissue is the standard of care for diagnosing sarcoma and ascertaining particular subtypes [4]. Although each subtype carries distinctive pathologic and biological features, the management of the majority of these tumors is usually similar. For localized STSs, a combination of surgery and radiation therapy is typically offered, while systemic therapy is the cornerstone treatment in a metastatic setting. Management of localized osteosarcoma and Ewing sarcoma requires the integration of multi-agent chemotherapy along with surgery and radiotherapy. Taken together, a multidisciplinary approach remains the most crucial element in optimizing the clinical outcomes of patients with sarcoma [1,2,5].

Although the majority of sarcoma cases are sporadic, some of these tumors are linked to familial and genetic predisposition phenotypes [1,2,6]. Examples of such associations between sarcomas and familial predisposition include malignant peripheral nerve sheath tumors (MPNSTs) occurring in individuals with type 1 neurofibromatosis, osteosarcomas and various STSs manifesting in patients with Li–Fraumeni syndrome, and a range of various sarcoma subtypes linked to tuberous sclerosis complex and hereditary retinoblastoma [7].

Data on germline genetic testing (GGT) for sarcomas are scarce, despite their increasing utility in enabling the selection of therapeutic options, cancer screening, and familial testing/counseling [8]. Moreover, the incidence and distribution of pathogenic germline variants (PGVs) differs according to geographic and ethnic background. Therefore, it is important to describe outcomes of GGT in diverse cohorts of patients to help guide personalized medicine [9].

Although GGT is recommended for specific subtypes of sarcomas, such as anaplastic rhabdomyosarcomas (RMSs), MPNSTs, and desmoid tumors, these recommendations are derived from few published series and institutional experiences [10,11,12]. Currently, there are no uniform guidelines to guide GGT in sarcoma patients [13].

In this prospective study, we investigated newly diagnosed adult patients with STSs and bone sarcomas at King Hussein Cancer Center (KHCC) to better understand the landscape of PGVs in our region and explore the potential actionability of these alterations.

## 2. Materials and Methods

### 2.1. Patients’ Population

After obtaining institutional Research Council (RC) and Institutional Review Board (IRB) approvals (IRB number 21 KHCC 27 and Clinicaltrial.gov identifier: NCT04920513), we launched this prospective study of GGT for adult patients who were diagnosed with STSs and bone sarcomas between March 2021 and October 2023. Eligible candidates were adults (>18 years) of Arab ancestry and were primarily treated at our hospital. All histopathology specimens were reviewed by a specialized sarcoma pathologist. Management plans were discussed at a sarcoma multidisciplinary clinic for all candidates. Eligible patients were evaluated at our clinical genetic counseling clinic. Patient and disease characteristics including age, gender, type of sarcoma, and location were collected. In addition, family history of malignancy was captured in first-, second-, and third- = degree relatives and a pedigree family tree was created. Of note, the presence of a second primary malignancy, whether synchronous or metachronous, was permitted and included in the analysis. Data were stored in a HIPAA-compliant database. 

### 2.2. Study Procedures

All newly diagnosed adult patients with STSs and bone sarcomas, regardless of the location, stage of the disease, and family history, were considered for enrollment in this study. Eligible candidates were approached by genetic counselors, study procedures were explained, and the consequences of genetic testing were thoroughly discussed. Consenting patients underwent GGT via an 84-gene multi-cancer panel (Invitae Corporation, San Francisco, CA, USA), Full-gene sequencing, deletion/duplication analysis, and variant interpretation were performed at Invitae Corp. as previously described [14] (Appendix A). Variants were interpreted using Sherloc, a refinement of the guidelines from the American College of Medical Genetics and Genomics classification, and pathogenic (P), likely pathogenic (LP), and variants of uncertain significance (VUS) were reported to clinicians [15,16]. Reports were sent to primary physician and genetic counselor, and then patients were informed regarding the results and their consequences.

### 2.3. Statistical Analysis

Descriptive statistics were applied when appropriate to report means, median, standard deviations, and proportions. The analysis included both P and LP variants and VUS. We examined for possible association of certain demographics and disease characteristics with P/LP variants’ detection rates. Such factors included gender, age, sarcoma subtype, grade, location, family history, and a history of non-cutaneous second primary cancers. The proportions of P/LP mutation in each comparison group were compared by Fisher’s exact test, with *p*-values < 0.05 indicating a statistically significant difference. All statistical analyses were performed using STATA version 18.

## 3. Results

### 3.1. Patients’ Characteristics

During the study period, a total of 87 patients with newly diagnosed sarcomas were eligible and consented for the study, whereas 21 candidates declined genetic testing and were not accrued in the study. Median age (range) was 48 (19–78) years, and 47 (54.0%) patients were females. Family history for malignancy was reported in 63 (72.4%) patients. Notably, 17 (19.5%) patients had a second primary malignancy other than sarcoma; mostly breast (*n* = 5), colorectal (*n* = 3), and ovarian (*n* = 2) cancer. Of the whole cohort, 41 (47.1%) presented with localized disease. The main histologic subtypes were gastrointestinal stromal tumor (GIST) (*n* = 12, 13.8%), liposarcoma (*n* = 10, 11.5%), and Ewing sarcoma (*n* = 10, 11.5%). The most frequent location was the extremities (*n* = 37, 42.5%), followed by retroperitoneal (*n* = 35, 40.2%), head and neck (*n* = 8, 9.2%), and chest (*n* = 7, 8.0%). Surgery was offered to 65 (74.7%) patients and radiation therapy to 33 (37.9%), while systemic chemotherapy was given to 50 (57.4%). Table 1 illustrates patient and disease characteristics.

### 3.2. Genetic Testing Results

A next-generation sequencing (NGS)-based 84-gene panel was performed on the 87 patients. Germline genetic alterations including both P/LP and VUS were detected in 58 (66.6%) candidates; 20 P/LP mutations were identified in 18 patients (20.7%); and 40 (45.9%) others had VUS. Figure 1 shows the distribution of P/LP and VUS mutations according to specific sarcoma pathology. The most frequently P/LP variants encountered were *APC* (*n* = 5, 22.7%), *TP53* (*n* = 3, 13.6%), *NF1* (*n* = 3, 13.6%), *BRCA2* (*n* = 2, 9.1%), *BRIP1* (*n* = 2, 9.1%), *MUTYH* (*n* = 2, 9.1%), and one (4.5%) patient each with *BRCA1*, *CDKN2A, CHEK2*, *PALB2*, *NTHL*. Notably, four out of the five *APC* P/LP mutations were increased risk allele (I1307K variant).

The rate of P/LP variants was higher among female patients (*n* = 13, 27.7%) compared to male patients (*n* = 5, 12.5%), (*p* = 0.042) and among patients aged 50 years or younger (*n* = 15, 26.8%) compared to older ones (*n* = 3, 9.7%) (*p* = 0.031), Table 2. Moreover, the presence of a second primary malignancy was associated with higher PGV frequency; 41.2% in patients with second primary cancer, compared to 15.7% in those without (*p* = 0.019). No other clinical characteristics were associated with PGV frequency (Table 2). Of note, results were also significant using the MULTTEST procedure with both Benjamini–Hochberg and False Discovery Rate. However, rates of VUS were significantly higher among older (>50 years) patients; 58.1% compared to 39.3% among younger ones, *p* = 0.047. Appendix A.

### 3.3. Actionable Genetic Alterations

P/LP variants in 8 (44.4%) patients (*BRCA1*, *BRCA 2*, *BRIP1*, *CDKN2A*, *CHEK2*, and *PALB2*) conferred potential actionability for therapeutic targets or clinical trials. Furthermore, all the detected P/LP variants had potential implications for cancer screening, family counseling, and cascade testing. Details are shown in Table 3.

### 3.4. Cascade Genetic Screening

Candidates who harbored P/LP mutations were offered cascade testing for their first-degree relatives. During the study period, 16 relatives of probands underwent cascade testing; 8 were relatives of one patient with *BRCA1*, and 4 (50.0%) were found to have the same *BRCA1* variant. Another relative, among two tested, of a patient with *MUTYH* had the same variant. Moreover, four family members of a patient with *APC* mutation underwent the test and one was positive. Accordingly, the total positive cascade testing rate was 6 (37.5%) of 16 relatives tested. Those relatives were counseled regarding the consequences of the P/LP alterations, and cancer screening imaging and exams were offered as well. Of note, although candidates were asymptomatic at time of testing, interestingly, the screening investigations revealed one ovarian, one breast, and one colorectal cancer.

## 4. Discussion

Sarcoma is a heterogeneous group of rare cancers originating from bone and other mesenchymal tissues. While the majority of sarcomas are sporadic, a subset of sarcomas have a well-established association with cancer predisposition syndromes [17,18]. Germline genetic testing has emerged as a tool that may help identify individuals at risk of developing sarcoma, and may carry implications for treatment and familial risk assessment. Germline testing can detect cancer predisposition genes which are directly linked to development of the disease and help understanding the biology and peculiar characteristics of each disease subtype [19,20,21,22,23].

Our study was a prospective genetic study of Jordanian patients newly diagnosed with both STSs and bone sarcomas. To the best of our knowledge, this study represents the first initiative to characterize the germline mutations of sarcoma patients from the Middle East region. We performed multi-gene panel testing to explore prevalence of specific gene mutations in our population. Around two thirds of our patients had genetic alterations, though the majority of these lacked clinical significance, and 20.7% were P/LP. Data on the sarcoma genomic landscape are still scarce, and current data are mainly derived from the Cancer Genomic Atlas and represent somatic, not germline, variants; around 25% of mutations occur in the cyclin-dependent kinase inhibitor pathway [24]. Nevertheless, in a more recent publication by Gounder et al. that reported on the genetic testing results of more than 7000 sarcoma patients treated at Memorial Sloan Kettering Cancer Center, most were in cell cycle regulator genes, such as *TP53*, *RB1*, and *CDKN2A* [25].

Genetic testing is one of the pillars of precision medicine. For sarcoma, germline testing enables physicians to identify hereditary cancer syndromes such as Li–Fraumeni and neurofibromatosis type-1. Early identification of various types of cancers, in addition to sarcomas, allows appropriate surveillance and proper family counseling, as many of these tumors present in the context of cancer predisposition syndrome [19]. In our cohort, 17 patients had a second primary malignancy and seven of them had pathogenic genetic mutations, with breast cancer as the most common. Notably, there was a statistically significant correlation between P/LP and presence of other primary malignancies. Moreover, we acknowledge that these P/LP mutations might be the driver for either of the cancers, or both. Consequently, genetic testing would be indicated in the international guidelines of each primary cancer.

Somatic mutation of *CDKN2A* has a known association with bone sarcomas, and in particular osteosarcoma; nevertheless, accumulating evidence has suggested that germline variants are linked with osteosarcoma as well. In this report, we described the *CDKN2A* c.172C > T nonsense (p.Gln58) LP variant in a 20 years old male with desmoplastic small round cell tumor, which is a rare soft tissue sarcoma typically diagnosed in males during the adolescent period [26]. Ferreira et al. explored genetic mutations that might have implications on the development of desmoplastic small round cell tumors, despite the rarity of the disease, and they described multiple somatic and germline mutations that were possibly related with this disease. Cell cycle-related mutations were one of the main groups of altered genes [27]. In addition to sarcoma, germline *CDKN2A* is involved with melanoma, pancreatic, and breast cancers. In this sense, *CDKN2A* can potentially be associated with a broad cancer predisposition phenotype.

Another important P/LP mutation is *NF1*. We identified three pathogenic alterations at Exons 9–35, Exon 13, and deletion in the entire coding sequence, in two patients with GIST and one with MPNST, consecutively. Individuals with NF1 have an increased lifetime risk of malignancy, and in particular of MPNST, GIST, and RMS [28,29].

MPNST in the background of NF1 is usually present at younger age and associated with worse disease outcomes when compared with sporadic counterparts [30]. On the other hand, patients with GIST and NF1 have distinctive patterns and the disease is usually multifocal, involves the small intestine, and expresses a low mitotic rate. Moreover, these tumors are frequently associated with proto-oncogene receptor tyrosine kinase (KIT) and discovered on the GIST-1 (DOG-1) protein. This fact carries clinical importance as this mutation is linked to imatinib resistance [31].

Another familial cancer predisposition syndrome is Li–Fraumeni syndrome, which is associated with germline mutation of *TP53*. The syndrome is typically related to breast cancer and bone and soft tissue sarcomas, in addition to other tumors such as suprarenal and neurological cancers [32]. We diagnosed three cases with *TP53* germline mutations, and all of these were detected in patients who had sarcoma only, with the exception of one case with sarcoma and breast cancer. Approximately one quarter of malignant tumors in *TP53* carriers are sarcomas, and the vast majority of these cancers are diagnosed at ages younger than 50 years. The most frequent sarcomas that are associated with Li–Fraumeni are osteosarcomas, undifferentiated pleomorphic sarcomas, and RMS [33]. Similar to neurofibromatosis, these patients usually present at a younger age. It is crucial to identify these patients, as patient education and counselling regarding cancer risks and early testing are paramount. Moreover, these mutations may harbor diagnostic and prognostic clues [34].

Another important hereditary cancer syndrome is Familial Adenomatous Polyposis (FAP). In this series, we reported four cases of *APC* increased risk allele; despite the fact that a missense variant in codon 1307 variant carries low penetrance, it increases risks for multiple cancers especially in Ashkenazi Jewish ancestry [35,36,37,38]. It seems our population might share similar risks; nevertheless, further studies are warranted to validate this finding.

*BRCA* germline mutations have low association with sarcomas; nevertheless, they have been implicated in RMS and uterine leiomyosarcoma [39,40]. Herein, we reported three pathogenic mutations, where two were *BRCA2* and one was *BRCA1*. However, two of these mutations were synchronous sarcomas and breast cancers and one case of breast sarcoma: in this context, these mutations might be the driver of breast cancer rather than sarcoma.

In recent years, genetic alterations have formed a target for novel agents. It is important to note that this applies primarily on somatic rather than germline mutations. In the era of molecular and genetic testing, it is estimated that 10–60% of primary sarcoma subtypes can be changed based on expert opinion using ancillary tools [41,42]. An important example of molecular positive impact in sarcoma is the targeting of c-kit mutations using imatinib and other anti-angiogenic therapies in GIST tumors [43]. Another example is the utilization of ribociclib and everolimus in patients with advanced dedifferentiated liposarcomas, targeting co-amplification of *CDK4* and *MDM2* genes [44]. Other examples include PARP inhibitor for homologous recombination repair gene mutation and pembrolizumab for mismatch repair-deficient cancers, which have been approved for other type of cancers [40,45,46]. Recently, newer classes of targeted and biological agents have emerged, with promising results in phase I and phase II clinical trials. Histone deacetylase 2 (HDCA2), a protein abnormally expressed in different types of cancer, forms a target for novel selective inhibitors through epigenetic modulation. Romidepsin is one example, which showed efficacy in dedifferentiated liposarcoma through suppression of (MDM2) expression and cell death enhancement [47,48], and Valproate is another inhibitor, which demonstrated some response in endometrial stromal sarcoma [49]. Similarly, Quisinostat showed promising results with SS18-SSX protein in synovial sarcoma [50]. Despite encouraging results, HDCA2 inhibitors still need extensive research to exhibit a breakthrough in sarcoma outcomes. In the same context, cancer stem cells (CSs) represent an emerging and hot topic for translational and clinical research. CSs have been correlated with carcinogenesis, chemo-resistance, disease progression, and metastasis. Epigenetic alterations play a crucial role in the pathophysiology of CSs, and CS markers such as CD133, ALDH, and PDGFRα were detected in various subtypes of sarcomas [51]. Novel drugs have proven some efficacy in sarcoma patients. Nilotinib is one example that inhibits MRP-1 and enhances anthracyclines’ effects [52]. Likewise, ruxolitinib, a JAK-STAT pathway inhibitor, displays a synergistic effect with chemotherapy. Nevertheless, the toxicity of these drugs remains the main obstacle facing these therapies [53].

In our series, we investigated patients and disease characteristics that might correlate with rates of germline mutations. We found younger age is associated with increased rates of P/LP; similarly, Vagher et al. found higher rates of mutations at younger ages in patients with STSs [13]. Female gender was also found to be associated with statistically significant higher rates of P/LP compared with male gender. Although a recent large review found male predilection in STSs and bone sarcomas, the results were inconsistent throughout different age and ethnic groups [54]. Nevertheless, data correlating sex and germline mutation in sarcoma are lacking, and our study provides a legitimate ground for future research. Hormonal effects on the genetic microenvironment and the influence of other clinical factors are still understudied. Similarly, we observed more P/LP in patients with family history for malignancy, but the difference was not statistically significant. This conclusion is parallel to Carvalho et al., who demonstrated higher mutation rates in patients with positive family history, although this was not statistically significant either [55].

Our study is a prospective cohort of germline testing for sarcoma patients. Data in the literature are limited and non-existent from this part of the world. The rates and types of genetic mutations were similar to that published in the international series. The number of our patients was relatively large, given the rarity of the disease. As aforementioned, the implications of germline testing are not only related to the patients themselves, but have consequences on patient cancer screening and cascade testing of their relatives. In our cohort, approximately one third of relatives who underwent testing were found to have P/LP mutations. This high percentage might be related to the small sample size and the large number of positive tests in one family. However, it is important to highlight the importance of cancer screening in this group, as we detected three cancers in patients who were completely asymptomatic.

Our study is not without limitations, as almost all patients enrolled were Jordanian and might not represent the region. The inclusion of other ethnicities or groups from the region would have provided better insight. Along with the relatively small number of patients, this would affect the generalizability of the study. Additionally, our study lacks clinical outcomes that can shed light on patients’ prognosis and treatment outcomes; this is a major caveat, because a correlation between genetic traits and clinical data is paramount to understand the implication of genetic alterations on patients’ outcomes with this relatively rare cancer. Moreover, somatic testing was not available, so we were unable to integrate germline and somatic tests. If available, it may enhance our understanding of sarcoma development and progression.

## 5. Conclusions

This is a prospective study of germline genetic testing for adult patients with soft tissue and bone sarcomas. Around two thirds of the patients were found to have germline genetic alterations and 20.7% of the cohort had P/LP germline variants. Younger age, presence of a second primary tumor, and female gender were significantly associated with higher P/LP rates. Many of the detected mutations were potentially actionable and almost all mutations had implications on cancer screening and family counselling. Larger studies and clinical data correlation are needed to validate our findings.

## Figures and Tables

**Figure 1 cancers-16-01668-f001:**
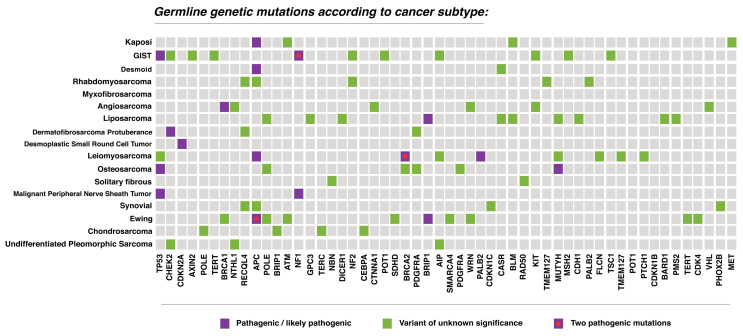
The distribution of P/LP and VUS mutations according to specific sarcoma pathology.

**Table 1 cancers-16-01668-t001:** Patients’ Characteristics (*n* = 87).

Characteristics		Number	(%)
Gender	Female	47	54.0
	Male	40	46.0
Age at diagnosis (years)	Median (Range)	48	
	≤50 years	56	64.4
	>50 Years	31	35.6
Positive Family History	Positive	63	72.4
	Negative	24	27.6
Sarcoma type	Soft Tissue Sarcoma	68	78.2
	Bone sarcoma	19	21.8
Histopathology Subtype	Gastrointestinal stromal tumor	12	13.8
	Liposarcoma	10	11.5
	Ewing sarcoma	10	11.5
	Leiomyosarcoma	7	8.0
	Undifferentiated pleomorhpic	7	8.0
	Osteosarcoma	7	8.0
	Dermatofibrosarcoma	5	5.7
	Synovial	4	4.5
	Chondrosarcoma	4	4.5
	Myxofibrosarcoma	3	3.5
	Angiosaromca	3	3.5
	Desmoid	3	3.5
	Rhabdomyosarcoma	3	3.5
	Malignant peripheral nerve sheath	3	3.5
	Kaposi	3	3.5
	Solitary fibrous	2	2.3
	Desmoplastic small round cell	1	1.2
Grade	1–2	49	56.3
	3	38	43.7
Location	Extremities	37	42.5
	Non-Extremities	50	57.5

**Table 2 cancers-16-01668-t002:** Rate of pathological/likely pathological and VUS variants.

Characteristics		N	Pathogenic/Likely Pathogenic *n* (%)	*p*-Value	VUS *n* (%)	*p*-Value
Total number		87	18 (20.7)		40 (46.0)	
Gender	Female	47	13 (27.7)	0.042	20 (42.6)	0.24
	Male	40	5 (12.5)		20 (50.0)	
Age at diagnosis (years)	Median (Range)	48 (19–78)				
	≤50	56	15 (26.8)	0.031	22 (39.3)	0.047
	>50	31	3 (9.7)		18 (58.1)	
Positive Family History	Positive	63	14 (22.2)	0.28	31 (49.2)	0.16
	Negative	24	4 (16.7)		9 (37.5)	
Histopathology	Soft tissue Sarcoma	68	14 (20.6)	0.48	28 (41.2)	0.046
	Bone sarcoma	19	4 (21.1)		12 (63.2)	
Grade	1–2	49	10 (20.4)	0.46	24 (48.9)	0.26
	3	38	8 (21.1)		16 (42.1)	
Location	Extremities	37	6 (16.2)	0.18	17 (45.9)	0.49
	Non-Extremities	50	12 (24.0)		23 (46.0)	
Second Primary Tumors	Yes	17	7 (41.2)	0.019	5 (29.4)	0.126
	No	70	11 (15.7)		35 (50.0)	

VUS: Variants of Uncertain Significance.

**Table 3 cancers-16-01668-t003:** Actionable genetic mutations.

Gene	Therapeutic Actionability	Cancer Screening and Counseling	Familial Syndrome
*APC*	-	Desmoid, familial adnomatous polyposis, adrenal hyperplasia, and other tumors, with cascade testing.	Familial Adenomatous Polyposis
*BRCA1*	olaparib + trabectedin	Breast, ovarian, prostate, pancreatic, and other cancer, with cascade testing.	
*BRCA2*	olaparib + trabectedin	Breast, ovarian, prostate, pancreatic, and other cancers, with cascade testing.	
*BRIP1*	NCT05787587	Ovarian cancer and breast cancers, with cascade testing.	
*CDKN2A*	palbociclib	Melanoma, pancreatic, neural system tumors, and other cancers, with cascade testing.	
*CHEK2*	NCT05252390/NCT04644068/NCT04550494/NCT05787587/NCT02401347/NCT06177171	Breast, colorectal, prostate, and other cancers, with cascade testing.	
*MUTYH*	-	Colorectal cancer, and extracolonic polyps and cancers, with cascade testing.	
*NF1*	-	Malignant peripheral nerve sheath tumors (MPNSTs), optic glioma, brain tumors, breast cancer, GIST, and adrenal gland tumors, with cascade testing.	Neurofibromatosis
*NTHL*	-	Colorectal, breast, endometrial, and duodenal cancer screening with cascade testing.	
*PALB2*	NCT05169437/NCT06177171/NCT05787587	Breast, ovarian, pancreatic, and other cancers, with cascade testing.	
*TP53*	-	Li–Fraumeni syndrome, including breast, sarcoma, brain, adrenocortical, skin, and other cancers, with cascade testing.	Li–Fraumeni

## Data Availability

Data used to generate this manuscript can be made available through the corresponding author upon reasonable request. All variants, including those reported in this manuscript, are submitted regularly to ClinVar. Our most recent submission has been processed in March 2023, which covered variants and interpretations to the end of 2022. Details are available at: https://www.ncbi.nlm.nih.gov/clinvar/submitters/500031 (accessed on 28 March 2023).

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
