# Peer review of "Germline Genetic Mutations in Adult Patients with Sarcoma: Insight into the Middle East Genetic Landscape"

_cancers, 2024, doi:10.3390/cancers16091668_

Round 1

Reviewer 1 Report

Comments and Suggestions for Authors

Dear editor,

Thank you so much for the opportunity to revise this manuscript entitled ‘’Germline Genetic Mutations in Adult Patients with Sarcoma: Insight into the Middle East Genetic Landscape’’. In this manuscript, the authors evaluated the prevalence of germline mutations in sarcoma patients, which had been treated in a tertiary center. While of interest, this manuscript should be improved before a potential publication.

Comments:

1. The subheads in the Abstract are unnecessary.

2. Statistical analysis methods should include the name of the statistical tests used for correlations.

3. Change Table-1/ Figure-1 by Table 1 or Figure 1.

4. Table 1: The total of the percentages by category should be 100% (e.g. female/ male rate sums 99.9%).

5. Genes should be written in italics.

6. Figure 1: Dismoid should be written desmoid.

7. Figure 1 should specify how many patients with a certain histologic subtype showed similar mutations on a particular gene.

8. The section 3.3 of results would benefit from a table with the potential clinical trials currently ongoing for the actionable mutations.

Comments on the Quality of English Language

Nothing to add. Minor english errors could be corrected.

Author Response

We would like to take this opportunity to thank you for the thoughtful and important comments, the points you raised would help us improve the quality of our manuscript. We have addressed reviewers’ comments one by one (below). You can find changes highlighted (underlined) throughout the manuscript.

Reviewer 1:

Thank you so much for the opportunity to revise this manuscript entitled ‘’Germline Genetic Mutations in Adult Patients with Sarcoma: Insight into the Middle East Genetic Landscape’’. In this manuscript, the authors evaluated the prevalence of germline mutations in sarcoma patients, which had been treated in a tertiary center. While of interest, this manuscript should be improved before a potential publication.

Comments:

  1. The subheads in the Abstract are unnecessary.

Thank you for your comment, modified as requested. (Lines 23/25/27/34)

  1. Statistical analysis methods should include the name of the statistical tests used for correlations.

Fisher’s exact test, due to small size population (some of the cells are < 5) (Line 108)

  1. Change Table-1/ Figure-1 by Table 1 or Figure 1.

Thank you for your comment, changed as requested. (Lines 124/132)

  1. Table 1: The total of the percentages by category should be 100% (e.g. female/ male rate sums 99.9%).

Thank you for your comment, modified as requested. (Table 1- row 3)

  1. Genes should be written in italics.

Genes have been changed to italic throughout the manuscript. (Lines 134/135/136/152/153/162/163/190/200/202/209/210/212/215/227/230/231/233/241/246/ 248/259)

  1. Figure 1: Dismoid should be written desmoid.

Thank you for your comment, typo has been corrected. (Figure 1)

  1. Figure 1 should specify how many patients with a certain histologic subtype showed similar mutations on a particular gene.

Appreciate your thoughtful comment. We had a maximum of 2 patients sharing the same specific, a new notation for duplicates is added (Figure 1)

  1. The section 3.3 of results would benefit from a table with the potential clinical trials currently ongoing for the actionable mutations.

Thank you for your valuable comment, table 3 is added (Line 155).

Comments on the Quality of English Language: Nothing to add. Minor English errors could be corrected.

Thank you for your comment, few typos were corrected (Lines 101/275 and figure 1) and table 2 last row, (font size)

Reviewer 2 Report

Comments and Suggestions for Authors

Overview

The present manuscript entitled "Germline Genetic Mutations in Adult Patients with Sarcoma: Insight into the Middle East Genetic Landscape" conducted a prospective genetic analysis on newly diagnosed sarcoma patients to understand the prevalence of pathogenic germline variants (PGVs) in the region and assess their potential actionability. The objectives and the rationale of the study are clearly stated. The figures and tables are adequate. The authors have stated the strengths and limitations of the current study. The structure and information flow make sense.

General comments and questions

The sample size of 87 patient and mutation information of only 84 genes significantly limit the generalizability of the findings, especially given the specific demographic focus.

The lack of clinical outcome data also limits the ability to understand the implications of detected PGVs on prognosis and treatment outcomes.

The statistical analysis did not control for multiple testing correction.

Author Response

We would like to take this opportunity to thank you for the thoughtful and important comments, the points you raised would help us improve the quality of our manuscript. We have addressed reviewers’ comments one by one (below). You can find changes highlighted (underlined) throughout the manuscript.

Reviewer 2:

The present manuscript entitled "Germline Genetic Mutations in Adult Patients with Sarcoma: Insight into the Middle East Genetic Landscape" conducted a prospective genetic analysis on newly diagnosed sarcoma patients to understand the prevalence of pathogenic germline variants (PGVs) in the region and assess their potential actionability. The objectives and the rationale of the study are clearly stated. The figures and tables are adequate. The authors have stated the strengths and limitations of the current study. The structure and information flow make sense.

Thank you for your review and comments  

General comments and questions

The sample size of 87 patient and mutation information of only 84 genes significantly limit the generalizability of the findings, especially given the specific demographic focus.

Appreciate your valuable comment. We totally agree that the number is relatively low, which may influence the generalizability of our results. But this is related to rarity of the disease and prospective nature of the study, our sample size is comparable with reports in the literature, and larger than many series. However, in regard to your comment, we have expanded this limitation in the discussion and conclusion parts (Lines 283/284/297/298). 

The lack of clinical outcome data also limits the ability to understand the implications of detected PGVs on prognosis and treatment outcomes.

Thank you for your comment, we agree that clinical data has an important role in understanding these genetic mutations. Nevertheless, such data requires years of follow up and authors might pursue this in the future. To address your comment, we have expanded this limitation in the discussion and conclusion parts (Lines 285/286/287/297/298). 

The statistical analysis did not control for multiple testing correction.

Thank you for your insight, of note, results were also significant using MULTTEST procedure with both Benjamini-Hochberg and False Discovery Rate (Lines 145/146).

Test

Raw (our p-value)

Hochberg

False Discovery

1

0.0420

0.0420

0.0420

2

0.0310

0.0420

0.0420

3

0.0190

0.0420

0.0420

We would like to thank reviewers for the valuable and thoughtful comments which will help improve our manuscript. We tried to address all your concerns and questions, hope you find our responses satisfactory.

Reviewer 3 Report

Comments and Suggestions for Authors

This is a well-written paper in terms of design, documentation etc. However, authors does not give a real specialized feedback to readers about Middle East Genetic Landscape.

Author Response

This is a well-written paper in terms of design, documentation etc. However, authors does not give a real specialized feedback to readers about Middle East Genetic Landscape.

Thank you for your comment, we acknowledged the limited scope of our study; including only Jordanians. Also this study lacks clinical data and somatic genes testing information. These points have been highlighted at the end of discussion, but in order to address your comment we expanded this limitation in the manuscript (Lines: 310-312). 

Reviewer 4 Report

Comments and Suggestions for Authors

This manuscript describes a prospective study of germline genomic testing for adult sarcoma patients in Jordan through use of an 84 gene Invitae panel test. The main strengths of this paper are the understudied population that it describes (sarcoma patients of Middle Eastern descent), and the number of patients studied (87), as well as the analysis and correlation between the rate of pathological/likely pathological variants and clinical characteristics. Specifically, in this study, younger age, female gender, and presence of a second primary tumor were significantly associated with higher P/LP rates. Information regarding cascade testing provided was also of interest.

General concept comments:

1) Please ensure in your introduction and conclusion that when speaking about lack of data on germline genetic testing in sarcoma patients that you are referring to this in adult patients, as there have been several studies of germline genetic testing in the pediatric cancer population that include sarcoma patients.

2) Please provide more detailed information regarding the specific Invitae panel used in the study. The manuscript states several times that an 84 gene panel was used, but provides no specifics on the genes included on this panel. Moreover, the reference provided (14, as listed on line 94) details information about a 29 gene panel specific to breast and ovarian cancer risk assessment.

3) Please explain the criteria for a positive family history of cancer (i.e.: 1st generation relative with cancer?)

4) Please provide more detailed demographic information in Table 1 regarding the names of all specific sarcoma diagnoses encompassed in the trial.

5) Please add additional data/figures in the results section that shows both genomic results and relevant clinical characteristics organized by individual patient for those with P/LP (would include VUS variants if possible, either in main figure or in supplementary figures).  I recommend an Oncoprint where each column represents an individual patient; this format could include rows for gender, second primary tumor, family history, etc, but ultimately defer to the authors to show this additional data in the manner that best illustrates it. Please include detailed information on each P/LP alteration (SNV details).

6) Please include additional information regarding patients/diagnoses in which no clinically significant findings were noted, as sometimes the absence of findings can be just as illuminating as those in which positive findings were noted. Inclusion of supplemental tables that detail information by individual for is recommended.

7) Please include information on whether any patients were treated with targeted therapies based on trial germline results. Additional outcome data, if available, would be of great interest.

8) CHEK2 is not associated with Li Fraumeni Syndrome, but rather its own cancer predisposition syndrome (PMID 37490054, PMID 37536919); please correct this in the conclusion.

9) Much of the discussion is devoted to explaining different types of cancer predisposition syndromes prior to detailing information about the specific cases from this trial. I recommend shortening these explanations, which could be done simply by adding a column in Table 3 that names the cancer predisposition syndrome associated with each gene. This table could then be referenced in the discussion where applicable, thereby negating the need to list out other associated malignancies in the text.

10) Please consider expanding your discussion regarding the clinical correlations noted in your cohort. For example, why do you think gender is associated with increased rates of germline alterations, and has this been seen in other studies/populations? Would you recommend germline testing for patients who present with another primary cancer given the findings in this study? Is younger age expected to be associated, as cancer predisposition patients are more likely to develop cancer at younger ages? Given that family history was not associated with P/LP variants, what does that mean for our historical reliance on this metric for referral to cancer geneticists and germline testing? Critically thinking about these issues and evaluating them in the discussion will make this a much more interesting paper.

Specific comments:

11) Please be consistent in the use of abbreviations (lines 19 and 20 reference PVGs rather than PGVs)

12) In Table 3, the heading "Genetic mutation" should be more accurately named "Gene"; also recommend including a column for the particular SNV noted.

13) Line 172-173: the association of some sarcomas with cancer predisposition syndromes has already been well-established

14) Lines 192-195: Recommend referring to Li Fraumeni and NF1 as hereditary cancer syndromes or cancer predisposition syndromes as they malignancies they are associated with are not limited to sarcomas (but rather can include them).

Comments on the Quality of English Language

Overall, the language quality is good. There are a few sentences that require rewording for clarity, but these issues are minor.

Author Response

General concept comments:

1. Please ensure in your introduction and conclusion that when speaking about lack of data on germline genetic testing in sarcoma patients that you are referring to this in adult patients, as there have been several studies of germline genetic testing in the pediatric cancer population that include sarcoma patients.

Thank you for comment, adults were added to introduction and conclusion as requested (underlined in lines: 24, 34, 72, 320)

2. Please provide more detailed information regarding the specific Invitae panel used in the study. The manuscript states several times that an 84 gene panel was used, but provides no specifics on the genes included on this panel. Moreover, the reference provided (14, as listed on line 94) details information about a 29 gene panel specific to breast and ovarian cancer risk assessment.

Thank you for your valuable comment, yes the 84 gene panel was used as stated in the manuscript. A full list of the genes panel has been added as requested in the supplementary table S1), with reference underlined (lines: 97, 98).

3. Please explain the criteria for a positive family history of cancer (i.e.: 1st generation relative with cancer?)

Thank you for your comment, positive family history was captured in 1st, 2nd and 3rd degree relatives regardless of generation. Added to manuscript (lines: 86).   

4. Please provide more detailed demographic information in Table 1 regarding the names of all specific sarcoma diagnoses encompassed in the trial.

All histologic subtypes were added to table 1; as requested.

5. Please add additional data/figures in the results section that shows both genomic results and relevant clinical characteristics organized by individual patient for those with P/LP (would include VUS variants if possible, either in main figure or in supplementary figures).  I recommend an Oncoprint where each column represents an individual patient; this format could include rows for gender, second primary tumor, family history, etc, but ultimately defer to the authors to show this additional data in the manner that best illustrates it. Please include detailed information on each P/LP alteration (SNV details).

A new table was added as requested (Supplementary table S2; referenced in line: 149)

6. Please include additional information regarding patients/diagnoses in which no clinically significant findings were noted, as sometimes the absence of findings can be just as illuminating as those in which positive findings were noted. Inclusion of supplemental tables that detail information by individual for is recommended.

All details, whether significant or not, are provided in Supplementary table S2.

7. Please include information on whether any patients were treated with targeted therapies based on trial germline results. Additional outcome data, if available, would be of great interest.

Thank you for your valuable comment, we agree on the importance of clinical outcomes. But unfortunately, such data is not available, and we addressed this in the limitations part at the end of discussion. In order to address your point, we expanded this limitation furthermore (lines: 313-315).

8. CHEK2 is not associated with Li Fraumeni Syndrome, but rather its own cancer predisposition syndrome (PMID 37490054, PMID 37536919); please correct this in the conclusion.

Thank you for your comment, corrected as requested (lines 231, 235).

9. Much of the discussion is devoted to explaining different types of cancer predisposition syndromes prior to detailing information about the specific cases from this trial. I recommend shortening these explanations, which could be done simply by adding a column in Table 3 that names the cancer predisposition syndrome associated with each gene. This table could then be referenced in the discussion where applicable, thereby negating the need to list out other associated malignancies in the text.

Your point is well taken, we added a new column in table 3 for familial syndromes as requested. And we tried to shorten familial syndromes part of discussion accordingly, with removing lines (218, 219, 220, 231, 232, 233, 245). However, we left some of the text because we feel it contains important findings or correlation with our results or may empower the scientific merit of our paper.

10. Please consider expanding your discussion regarding the clinical correlations noted in your cohort. For example, why do you think gender is associated with increased rates of germline alterations, and has this been seen in other studies/populations? Would you recommend germline testing for patients who present with another primary cancer given the findings in this study? Is younger age expected to be associated, as cancer predisposition patients are more likely to develop cancer at younger ages? Given that family history was not associated with P/LP variants, what does that mean for our historical reliance on this metric for referral to cancer geneticists and germline testing? Critically thinking about these issues and evaluating them in the discussion will make this a much more interesting paper.

Thank you for your thoughtful comment, we have addressed the clinical data at the end of discussion. But as you mentioned further discussion would improve our manuscript. In order to address the points you highlighted we expanded that part as follow:

  • For gender association, we cited a recent review addressing sex predilection in sarcoma. However, to the best of our knowledge, no existent data investigated the correlation between gender and germline mutations in sarcoma. We think this is one of the strength of our study, and further studies are warranted to study this correlation; added to discussion part (lines: 289-294).
  • As regard genetic testing for patients with second primary, thank you for raising this really interesting point, but we think the recommendation for genetic testing would be outlined by international guidelines, because such a decision warrants epidemiologic and financial studies. We added this to the discussion part (lines: 202-203).
  • Although P/LP mutations are associated with younger age, we cannot confirm the fact that those patients developed cancer at younger age than other patients in our population. Another epidemiology study is required, in order to calculate the median age at diagnosis in the whole population and compare it with our findings, but unfortunately this data is not available.
  • With regard family history, as stated in the discussion, family history was associated with higher rates of P/LP 14 vs 4, but it was not statistically significant. This may be related to the relatively small sample size and the fact that our study was not powered to detect the difference. This was addressed in the limitations part. Regarding referral for genetic testing, again we think international guidelines are the best to dictate clinicians practice in this regards.

Specific comments:

11. Please be consistent in the use of abbreviations (lines 19 and 20 reference PVGs rather than PGVs)

Thank you for your comment, corrected as requested.

12. In Table 3, the heading "Genetic mutation" should be more accurately named "Gene"; also recommend including a column for the particular SNV noted.

Well noted, the heading changed as requested. Details of genes (including SNV) are provided in the new Supplementary table S2.

13. Line 172-173: the association of some sarcomas with cancer predisposition syndromes has already been well-established

Thank you for your comment, corrected as requested (Lines: 174-175).

14. Lines 192-195: Recommend referring to Li Fraumeni and NF1 as hereditary cancer syndromes or cancer predisposition syndromes as they malignancies they are associated with are not limited to sarcomas (but rather can include them).

Well noted, changed in in the discussion (Lines: 194-196).

Comments on the Quality of English Language

Overall, the language quality is good. There are a few sentences that require rewording for clarity, but these issues are minor.

Typo: P/PL (Line: 201) Corrected.

Line (255,256): the genetic alterations have become targetable with agents developed for the same cancer subtype or other cancer types, changed to: ”formed a target for novel agents”.

Line (286-287): this aligned with Vagher, et al. finding of increased mutation prevalence at younger age in patients with STS. Changed to: “,similarly, Vagher et al. found higher rates of mutations at younger age in patients with STS”

Round 2

Reviewer 3 Report

Comments and Suggestions for Authors

The main problem is still there, that real conclusions were not drawn from these data by authors.

Author Response

Thank you for your comment, the conclusion part in the abstract (Lines: 38-44) is changes to:” Almost one in five adult patients with soft tissue and bone sarcomas harbor pathogenic or likely pathogenic variants. Many of these variants are potentially actionable, and almost all have implications on cancer screening and family counselling. In this cohort from the Middle East, younger age, presence of second primary tumor and female gender, were significantly associated with higher pathogenic or likely pathogenic rates. Larger studies able to correlate treatment outcomes with genetic variants are highly needed.”

Hope you find our response satisfactory.

Reviewer 4 Report

Comments and Suggestions for Authors

Thank you for making the suggested adjustments and for your comments.

Supplemental Table S2: Recommend removing location column and adding a column detailing the diagnosis of the primary tumor. The location column in its current form only states lower vs upper limb, but in the manuscript there are details about head/neck, torso, etc.; regardless, I think this info is less important and interesting compared to knowing what the primary tumor type is.

Comments on the Quality of English Language

Overall, the language quality is good. There are a few sentences that require rewording for clarity, but these issues are minor.

Author Response

Thank you for your comment, we changed location to tumor diagnosis as requested
